# Development of an Internet of Things-Based Ultra-Pure Water Quality Monitoring System

**DOI:** 10.3390/s25041186

**Published:** 2025-02-15

**Authors:** Mehmet Akif Öztürk, Emre Ünsal, Ahmet Fırat Yelkuvan

**Affiliations:** 1Graduate School of Natural and Applied Sciences, Sivas Cumhuriyet University, Sivas 58140, Turkey; m.akifozturk95@gmail.com; 2Department of Software Engineering, Faculty of Technology, Sivas Cumhuriyet University, Sivas 58140, Turkey; 3Department of Computer Engineering, Faculty of Engineering, Sivas Cumhuriyet University, Sivas 58140, Turkey; aftyelkuvan@cumhuriyet.edu.tr

**Keywords:** internet of things, embedded systems, water quality monitoring, water conductivity sensor

## Abstract

Monitoring ultra-pure water quality is crucial in dialysis centers and medical laboratories as even minor impurities can significantly impact health and diagnostic accuracy. In addition, the semiconductor industry needs and uses a significant amount of ultra-pure water. This study introduces an Internet of Things-based system for real-time monitoring and analysis of ultra-pure water conductivity, temperature, and other key parameters. The proposed system integrates a high-precision conductivity sensor, an ESP32 microcontroller, and a web-based interface to enable remote data access and visualization. Data transmission is through wireless communication, and values are stored on a web-based server for long-term analysis. Rigorous tests conducted at Sivas Numune Hospital validated the system’s reliability, accuracy, and ability to maintain stringent ultra-pure water quality standards. This robust and cost-effective monitoring solution addresses the limitations of conventional systems and provides real-time insights, ensuring consistent water quality for sensitive medical applications.

## 1. Introduction

Internet of Things (IoT) brought about major changes in various fields by enabling devices to collect, share, and analyze data through interconnected networks [1,2]. This technological advance is widely applied in important sectors such as healthcare [3,4,5], agriculture [6,7,8], manufacturing [9], and environmental monitoring [10]. In the context of water quality monitoring, IoT provides real-time insights that are crucial to health, safety, and operational efficiency [11,12].

Pure water quality monitoring, in particular, has emerged as an essential function due to its significant role in medical procedures, laboratory testing accuracy, and public health [12]. Moreover, ultra-pure water has significant applications in the semiconductor industry, being essential for the manufacturing of processors and microchips [13,14]. In these processes, ultra-pure water is utilized as a cleaner, reagent, solvent, and a key component in various chemical mixtures [15,16].

The quality of the water is determined by analyzing various parameters, including bacterial content, pH level, chemical composition, turbidity, and conductivity [11,17,18,19,20]. These factors are crucial indicators of water purity, as contaminants such as dissolved minerals and microorganisms can significantly compromise its quality. Among these indicators, conductivity is particularly important, as it reflects the concentration of electrolytes, including salts, acids, and bases, in the water. This parameter provides valuable insights into the chemical and ionic characteristics of the water, enabling the identification of potential pollutants and the overall assessment of water quality [21,22]. This study aims to address the challenges in monitoring ultra-pure water quality by developing an IoT-based system that measures parameters such as conductivity, temperature, water flow, and consumption in real time. The proposed system integrates advanced embedded technologies and web-based data management for improved monitoring and analysis.

Compared to drinking water in ultra-pure water systems, the electrolytic conductivity and resistivity of purified water is a key indicator of ionic contamination. Conductivity is measured in microsiemens per centimeter (μS/cm), common in the pharmaceutical and power industries, while resistivity, its reciprocal, is measured in megohm-centimeters (MΩ·cm), typical in microelectronics. At 25 °C, ultra-pure water has a conductivity of 0.05501 μS/cm and a resistivity of 18.18 MΩ·cm. Even minimal contamination, such as 0.1 ppb of sodium chloride, can increase conductivity to 0.05523 μS/cm and decrease resistivity to 18.11 MΩ·cm [23]. Ultra-pure water is highly susceptible to contamination by carbon dioxide, which forms conductive carbonic acid. To prevent this, conductivity probes, which combine conductivity and temperature sensors for accurate temperature compensation, are installed directly in the system piping for continuous, real-time monitoring [24].

Recent years saw an expansion of research on IoT-based water quality monitoring systems. Malissovas et al. [25] proposed a real-time smart water quality monitoring system equipped with a silicon-based sensor which measures the salinity, acidity, and temperature of the water. Another proposed work focused on an IoT-based platform to assess drinking water quality monitoring in real time [26]. The system continuously monitors the water quality parameters using multiple sensors and transmits the data to a cloud-based platform for real-time analysis. Chafa et al. [27] proposed a real-time IoT-based water quality monitoring and control system for water treatment plants. The system integrates sensors with a microcontroller to monitor deviations, triggering autonomous control measures when necessary. Another IoT-based water quality monitoring system for a pond is developed for the fish farming industry. The system monitors essential parameters such as pH, electrical conductivity, dissolved oxygen, and turbidity. The collected data are displayed on a dashboard, enabling farmers to monitor water conditions in real time and implement necessary measures to maintain optimal quality [28]. An autonomous IoT-driven real-time monitoring system is developed for industrial wastewater management. This system incorporates multi-parameter sensors to measure key water quality indicators, including pH, dissolved oxygen, electrical conductivity, total dissolved solids, and turbidity. By leveraging machine learning for predictive analytics, the system facilitates timely interventions and ensures compliance with environmental regulations [29]. Saravanan et al. [24] developed a system integrated with IoT to enhance real-time water quality monitoring. The system captures various water parameters such as temperature, flow, and color, and detects contamination of water and pipeline leakage with the help of cloud-based storage and wireless communication. These studies highlight the potential of IoT in providing real-time, accurate, and scalable solutions for monitoring water quality, although they often lack adaptability for highly specialized contexts such as ultra-pure water monitoring.

Despite the fact that these studies highlight the potential of IoT in providing real-time, accurate, and scalable solutions to monitor water quality, their scope of application often remains broad and generalized. For example, existing systems typically focus on parameters such as pH, turbidity, and dissolved oxygen in drinking water or natural water bodies. However, these solutions lack the precision and adaptability required for highly specialized contexts, such as ultra-pure water monitoring. Moreover, most systems prioritize cost-efficiency and general applicability over the high precision needed for medical or laboratory-grade water monitoring.

In contrast, the present study targets the specific challenges of monitoring ultra-pure water quality, which is crucial in applications like dialysis centers, medical laboratories, and also semiconductor manufacturers. The applications in these environments are extremely sensitive to impurities, as even minor deviations in water quality can adversely affect patient health or the accuracy of diagnostic results. Unlike conventional systems, our study emphasizes the monitoring of electrical conductivity as a primary parameter to assess water purity, integrating a high-precision sensor system and embedded technologies for real-time analysis. The data collected by the system are stored on a cloud-based platform, allowing remote access and long-term trend evaluation.

By addressing the gaps in adaptability and precision observed in the existing literature, this study aims to provide a robust and reliable solution specifically tailored for ultra-pure water quality monitoring. The integration of advanced IoT frameworks and embedded technologies ensure high sensitivity, accuracy, and real-time monitoring, setting the proposed system apart from previous approaches.

## 2. Materials and Methods

This section provides details on embedded system design, sensor manufacturing, and embedded system software development. It also elaborates on the process of integrating sensor data into embedded systems and highlights advancements in monitoring and reporting these data through developed web-based software.

The most important parameter to examine for determining the quality of ultra-pure water is its electrical conductivity. The concepts of conductivity and resistance are inversely proportional to each other. The resistivity (*p*) of the material or liquid, measured in ohm-meters (Ω·m), is defined as the resistance of a unit cube with perfectly conductive contacts on opposite surfaces [30]. For materials with other geometries, the resistance (*R*) value can be calculated using Equation (Equation 1).(1)$$R=\frac{pL}{A}$$

*L* represents the length of the sensing surface where the resistance measurement is taken, while *A* denotes the cross-sectional area of the material’s contact surfaces. The resistance value is measured in ohms (Ω). A material with a resistivity of 1 Ω exhibits a resistance of 1 Ω when measured across opposite faces of a 1 cm × 1 cm × 1 cm cube. Measured conductance (*Y*), on the other hand, is the reciprocal of resistance (*R*) and can be calculated using Equation (Equation 2).(2)$$Y=\frac{1}{R}$$

In this paper, $Y_{x}$ represents the conductivity measured in μS/cm. The unit of conductance (*Y*) is known as Siemens (*S*), and the conductivity ($Y_{x}$) measurements are expressed in subunits such as S/cm, mS/cm, or μS/cm.

The components of the embedded system developed for the IoT-based ultra-pure water monitoring system, along with the system block diagram, are presented in Figure 1.

The hardware design of the embedded system is based primarily on the ESP32 microcontroller module [31]. Various details regarding the ESP32 microcontroller and the other hardware components are presented below.

### 2.1. Development of the Embedded System Board

#### 2.1.1. Microcontroller Selection

The ESP32 microcontroller was chosen as the core processing unit for the system due to its high performance, versatility, and support for IoT applications. It features a dual-core Tensilica LX6 processor with a clock speed of up to 240 MHz, integrated Wi-Fi and Bluetooth capabilities, and multiple general purpose input/output (GPIO) pins for interfacing with external peripherals [31]. These features make the ESP32 ideal for low-cost, low-power, and real-time embedded applications. The specifications of the ESP-32 microcontroller are provided in Table 1.

#### 2.1.2. Signal Conditioning and Conversion

The system incorporates an ADS1115 analog-to-digital converter (ADC) to accurately measure the conductivity and temperature of water. This 16-bit ADC provides high resolution and precision, converting the analog signals from the sensors into digital data for further processing [32]. The ADS1115 was chosen for its programmable gain amplifier and wide input voltage range, which ensure accurate readings across various conditions.

In addition, the operational amplifier (op-amp) TL064 is used for signal amplification. This low-power JFET-input op-amp is highly suitable for applications requiring low noise and high input impedance, making it a great choice for the signal conditioning circuit in this system [33].

#### 2.1.3. Multiplexer and Demultiplexer

The HC4053 multiplexer/demultiplexer is includes to handle multiple analog signals from the sensors and route them efficiently to the ADC and processing unit [34]. Its complementary metal-oxide semiconductor (CMOS)-based design ensures low power consumption and high-speed operation, which are essential for maintaining the system’s efficiency and responsiveness.

#### 2.1.4. Integrated Synchronous Voltage Regulator Circuit

MP2307 is used to regulate the power supply for the embedded system, providing stable +5 V and −5 V outputs required for various components such as the microcontroller and signal conditioning circuits [35]. Its low power loss and compact design make it ideal for ensuring energy efficiency and reliable operation in portable and IoT-based systems. MP2307 also features built-in protection mechanisms, including overcurrent and overvoltage protection, and thermal shutdown, ensuring system safety and durability.

### 2.2. Development of the Conductivity Sensor

The design of the sensor probe is critical for ultra-pure water quality monitoring system to ensure the accuracy and reliability of the measurements. The sensor probe is specifically designed to measure the conductivity and temperature of ultra-pure water, which are key indicators of ultra-pure water quality.

The sensor probe consists of two main components: Conductivity Measurement Electrodes and the Integrated Temperature Sensor. The probe includes two electrodes made of corrosion-resistant materials, such as 314 stainless steel or titanium, to ensure durability and consistent performance in water environments. These electrodes are precisely spaced to maintain a known cell constant, which is crucial for accurate conductivity measurements [30]. The temperature sensor in the form of a negative temperature coefficient (NTC) thermistor is embedded in the probe to measure water temperature. This is necessary because water’s conductivity is temperature-dependent, and the measured conductivity values must be corrected to a reference temperature of 25 °C.

An alternating current (AC) signal in the range of 1–10 kHz is utilized to account for the effect of dissolved substances on conductivity. This approach enables accurate measurements without risk of electrolysis. Conductivity exhibits a significant temperature coefficient in liquids, with changes in temperature substantially affecting the readings. For ultra-pure water, a temperature change of 1 °C can result in up to 4% increase in conductivity [23]. This underscores the importance of incorporating temperature compensation in conductivity measurement systems to ensure precise and reliable results.

When measuring a 1 μS/cm solution, the cell is configured with large-area electrodes and a narrow gap to optimize accuracy. For instance, a cell with a constant of 0.01/cm typically exhibits a measured resistance of approximately 10 kΩ instead of 1 MΩ. Measuring 10 kΩ is significantly easier and more reliable than measuring 1 MΩ, which often encounters higher noise and precision challenges. Consequently, the measurement device is designed to operate across a range of cell constants, enabling it to accommodate both ultra-pure water and highly conductive solutions such as seawater while maintaining the same effective resistance range. Such adaptability ensures consistent and accurate measurements across varying conductivity levels. The conductivity cell diagram illustrates the arrangement of electrodes, the current flow, and the voltage applied, which are essential for determining the conductivity of the water as sample shown in Figure 2.

The cell constant *K* is the ratio of the distance between the electrodes to their surface area.(3)$$K=\frac{L}{A}$$
where *L* represents the distance between the electrodes and *A* is the surface area of the electrodes. Using this cell constant, the electronic circuit calculates the conductance (*Y*) of the liquid. The system measures the voltage (*V*) and current (*I*) to determine the resistance (*R*). Conductance is then derived as the reciprocal of resistance, which can be expressed as(4)$$Y=\frac{I}{V}$$

This relationship between cell constant and conductance values allows the system to accurately compute the conductivity of the liquid ($Y_{x}$), enabling precise monitoring of the water quality, as given in Equation (Equation 5).(5)$$Y_{x}=KxY$$

Although the formula is highly applicable, even minor imperfections on the surface of the sensor material can alter the electrode surface area (*A*), potentially affecting the measurement accuracy. To mitigate this, it is crucial to ensure that the sensor material is processed with precision, maintaining a smooth and consistent surface. In addition, proper insulation of the electrodes is essential to prevent external interference. The proximity of the temperature sensor to the liquid is also of great importance, as it ensures accurate temperature readings, which are critical for compensating conductivity measurements. This meticulous approach to design and manufacturing is essential to ensure the reliability and precision of the sensor system.

The test sensor probe used in this study was made of polished 314-grade stainless steel for higher durability and resistance to corrosion. The sensor probe is connected with the water purification system, using coupling components made of Polytetrafluoroethylene (PTFE) Teflon sheets. These components were precisely crafted on lathe and milling machines.

The components of the sensor probe, along with all connection parts, are shown in Figure 3. As shown in the figure, a cylindrical shape was chosen to make production easier. To ensure uniformity in the surface area of the electrode across all sensors, each unit was tested and refined prior to the application of waterproof resin. The temperature sensor was strategically placed within the inner electrode, insulated, and placed as close as possible to the liquid medium to guarantee precise temperature measurements. The result of such a meticulous design and manufacturing process is a highly reliable and accurate sensor probe.

### 2.3. Flow and Water Pressure Sensors

The YF-S201 model water flow sensor (Figure 4a) is designed to measure the flow rate of water and can be used in projects that want to monitor water flow [36]. With an operating voltage of 5 V–24 V, an operating temperature range of −20 °C to 85 °C, and an operating current of 15 mA, this sensor can measure water flow of 1–30 L per min. Normal working pressure is specified as ≤1.75 MPa.

The Gravity Water Pressure Sensor (Figure 4b) is a sensor that can measure pressure from 0 to 1.6 MPa and output the measurement results as a linear voltage output from 0.5 V to 4.5 V [37]. This sensor is particularly suitable for use in projects where water pressure needs to be controlled. With an operating voltage of 5 V, an operating temperature range of −20 °C to 85 °C, a response time of <2 ms, and an operating current of 2.8 mA, this sensor measures with an accuracy of 0.5% to 1%. It is IP68 waterproof and outputs 0.5 V at 0 MPa pressure and 4.5 V at 1.6 MPa pressure.

### 2.4. Working Principles of the Electronic Circuit

The circuit illustrated in Figure 1 is based on the ESP32 microcontroller and aims to provide precise measurements suitable for industrial applications while maintaining internet connectivity. The system is equipped with additional hardware to digitize pressure and flow sensors, enabling continuous monitoring and control of the pure water production system.

The power supply for the circuit utilizes MP2307 regulators (Figure 5a,b) to generate +5 V, GND, and −5 V.

To achieve symmetric power, 12 V or higher input voltage is processed through two synchronous regulators, as depicted in Figure 5. The system is fine-tuned using adjustable resistors (R4 and R6) to minimize offset values. The current from the sensor probe is routed through the SEN1_IN line into the TL064 (Figure 5b) operational amplifier, where it is converted to a voltage signal suitable for the ADC. The reference resistor R13, with a value of 1 kΩ, supports this conversion. Calculations for signal values are provided in Section 3.

Figure 6 presents the circuit design for processing signals from the conductivity and temperature sensors. A 1 kHz PWM signal from the microcontroller is amplified using the 74HC4053 multiplexer/demultiplexer (Figure 6c) to generate the excitation signal, labeled as SEN1_OUT. To limit current flow and to prevent short circuits, a 1 kΩ resistor is connected at the SEN1_OUT square wave output, restricting the maximum current to 20 mA.

As shown in Figure 6b, the SEN1_IN signal from the sensor output is applied to channel 1 of the TL064 JFET op-amp. Resistor R13 determines the magnitude of the SEN1_IN signal applied to the input of the op-amp depending on the fluid resistance, resulting in an amplified SEN1+ signal at the output. The SEN1+ signal is then inverted on the second channel of the TL064 to produce the SEN1− signal. The SEN1+ and SEN1− signals are applied to the 0Y and 0Z inputs of the HC4053 mux/demux for rectification (Figure 6c). As a result, the signal from the sensor is first rectified, then the rectified signal is filtered to remove noise and the SEN1_ADC signal is obtained.

The SEN1_ADC signal is transmitted to the AIN0 input of ADS1115 and digitized with 16 bit resolution. The digitized conductivity value is transmitted to the MCU via the $I^{2}C$ communication interface. This design allows the system to measure conductivity levels from 0.001 μS to 20 μS, providing an ideal capability for monitoring ultra-pure water with a minimum conductivity value of 0.055 μS. The MCU also retrieves temperature data from a 10 kΩ nominal NTC sensor embedded in the probe core for accurate liquid temperature measurement.

Before obtaining the SEN1_ADC output of the TC4053, a low-pass filter is formed using an inductor and a capacitor to improve the accuracy and stability of the signal (Figure 6c). This filter eliminates high-frequency noise in the signal, preventing it from reaching SEN1_ADC. Additionally, it reduces noise caused by electromagnetic interference and power supply fluctuations, ensuring that a cleaner and more reliable signal is delivered to the ADC input. Thus, the accuracy and stability of the analog signals are enhanced before digital processing.

Figure 7 illustrates the ESP32 microcontroller and its connections to the peripherals. Flash memory (Figure 7d) is used to provide the ESP32 with an extra 16MB of storage and communicates externally via the SPI bus. The CP2102 UART/USB bridge (Figure 7c) provides USB interface communication for programming and debug operations. A 2 × 16 liquid-crystal display (LCD) screen (Figure 7b) is used as user information display. The communication between ADS1115 and ESP32 is provided via SDA and SCL pins. The AC_PWM signal is used to change the frequency of the excitation signal. A buzzer is used as a warning and alarm indicator in the circuit.

### 2.5. Prototyping and Testing

The designed circuit components were assembled on a perforated prototyping board shown in Figure 8 to evaluate the functionality and accuracy of the ADC readings and calculations. This preliminary testing phase ensured that all electronic components operated as expected and provided reliable data. After verifying the circuit’s performance, it was enclosed within a protective printed circuit board (PCB) casing for enhanced safety and stability. In addition, an LCD screen was integrated into the setup to display real-time system parameters, ensuring that the testing process was secure and user-friendly. These measures facilitated an accurate assessment of circuit components while safeguarding the system during further experiments.

The electronic circuit for conductivity measurement provides precise generation and handling of four critical signals: SEN1_OUT, SEN1_IN, AC_PWM, and AC_EN1. The reference signal SEN1_IN and the output signal SEN1_OUT is given in Figure 9a and Figure 9b, respectively.

AC_EN1 Signal: This signal protects the sensor from oxidation when the system is not actively measuring. To achieve this, the signal remains at 1 (Logic High) during active use and switches to 0 (Logic Low) during idle periods. This mechanism helps extend the sensor’s operational lifespan.AC_PWM Signal: This signal generates the required frequency for the system. Dissolution of oxygen and hydrogen ions in the liquid can alter conductivity and lead to measurement inaccuracies. To mitigate this, tests were conducted, and a minimum PWM signal frequency of 1000 Hz was found to be optimal. The reference signal used for measurement is shown in Figure 9a.

The excitation signal changes the current value depending on the resistance of the fluid as it passes through the conductivity sensor. This current is converted into a voltage signal which can be read by the ADC using the TL064 operational amplifier. The output signal observed during the process is shown in Figure 9b. This rectified current signal is then filtered to remove noise and transmitted to the ADC for further processing. This approach ensures accurate and dependable conductivity measurements, with signal processing and filtration enhancing the system’s overall reliability and precision.

The inverting amplifier circuit shown in Figure 10 is used to facilitate the accurate determination of unknown $R_{FLUID}$ resistance. The circuit operates on the following equation:(6)$$V_{OUT}=-\frac{V_{IN}*R_{REF}}{R_{FLUID}}$$

In this equation,
$R_{REF}$ represents the known reference resistance.$V_{IN}$ represents to the excitation signal applied to the sensor.$R_{FLUID}$ represents the resistance of the liquid being measured.$V_{OUT}$ represents voltage of the amplifier circuit.

Based on Equation (Equation 6), the following equation is formulated to determine $R_{FLUID}$:(7)$$R_{FLUID}=-\frac{V_{IN}*R_{REF}}{V_{OUT}}$$

Equation (Equation 7) enables for the precise calculation of the liquid’s resistance, which is directly used to determine its conductivity. The resistance values thus found are then converted into conductivity using the pre-calibrated cell constant, ensuring accurate results.

### 2.6. Development of the Embedded System Software

The embedded system software is designed to enable seamless integration of hardware components, real-time data processing, and efficient communication with the web platform. To achieve real-time operation, the software is developed using C/C++ programming languages within a Real-Time Operating System (RTOS) framework. This structure ensures the concurrent execution of multiple tasks and the efficient handling of critical processes.

As shown in Figure 11, the embedded system software utilizes an RTOS framework to manage and execute multiple tasks concurrently, ensuring real-time functionality and synchronization. At the heart of the software is the System Initialization process, which runs during startup to configure the microcontroller. This process includes setting up input and output ports, initializing variables, assigning default values, and reading configuration data from EEPROM. Furthermore, it configures essential operational parameters, such as the microcontroller’s clock frequency, ensuring stable and efficient system performance.

A key component of the software is the sensor task management, which handles gathering data from the conductivity and temperature sensors. The output sensor data are captured through the ADC, where it is processed, calibrated, and validated for errors. These steps ensure accurate and reliable measurements essential for ultra-pure water quality monitoring.

The Wi-Fi and IoT tasks enable real-time data communication that manages interactions with the server. These tasks are crucial for establishing and maintaining a stable network connection, transmitting sensor data at predefined intervals, and handling any communication errors. The real time data exchange enables remote monitoring and ensures that essential information is always accessible.

Another important part of the software is the Human–Machine Interface (HMI) task, which drives the LCD display to present real-time information to the user. Key parameters such as conductivity, temperature, water flow, and system status are displayed clearly, providing immediate insights into system performance. The HMI task also handles user inputs and manages visual and auditory notifications of any anomalies.

The RTOS framework orchestrates these tasks, enabling them to operate in parallel while maintaining synchronization and real-time responsiveness. This integrated and modular approach provides a robust and reliable software platform, meeting the stringent requirements for a real-time ultra-pure water quality monitoring system.

The embedded system software for the IoT-based ultra-pure water quality monitoring system was developed using robust tools and frameworks to ensure high reliability, efficiency, and scalability.

The embedded software was developed on PlatformIO IDE with the ESP32’s ESP-IDF framework for efficient embedded programming in C/C++. The system was designed to leverage the RTOS capabilities of the ESP32, enabling precise task scheduling and concurrent execution of key functionalities. JTAG and Serial Monitoring tools were employed for debugging and performance optimization. Wi-Fi and JSON libraries were integrated to enable seamless communication and web connectivity. Additionally, drivers for the 2 × 16 LCD display and control button modules were implemented for data visualization and user interaction.

### 2.7. Development of the Web Application

The PHP and Laravel frameworks are used to develop web applications and IoT-based services for an ultra-pure water monitoring system. The embedded system collects real-time sensor data, such as conductivity, temperature, and other relevant parameters, and formats it into structured JSON packets. Data from IoT devices, transmitted in JSON format, are parsed through a web service and saved into a MySQL database. These packets are transmitted to the web server via HTTP POST requests, where the data are processed and stored for long-term analysis and visualization. The system diagram is shown in Figure 12.

The web server supports remote access via a web-based interface and mobile applications providing users with access to real-time data, allowing them to set alert thresholds and analyze historical trends. This IoT communication framework is designed to be user-friendly, scalable, and capable of ensuring the continuous monitoring of water quality parameters.

The real-time sensor values obtained with the IoT devices, as displayed on the developed web-based monitoring system, are presented in Figure 13a. The sensor values summary screen, shown in Figure 13b, provides a summary of sensor measurements recorded at 1 h intervals, offering a detailed view of the system’s historical data collection capabilities.

## 3. Experimental Work

The experimental work aims to validate the reliability and robustness of the developed IoT-based ultra-pure water quality monitoring system. The accuracy of the key parameters such as conductivity and temperature was evaluated by the existing local conductivity sensor. The communication framework designed with stable and reliable data transmission to the web server in mind. All tests were conducted under real-world conditions to ensure the system’s effectiveness and suitability for practical deployment.

### 3.1. Testing the Prototype

The first prototype of the system was installed on the ultra-pure water supply system in the biochemistry laboratory at Sivas Numune Hospital for testing. As shown in Figure 14b, the prototype board is able to produce consistent and reliable data when compared with a widely used sensitive conductivity meter (Figure 14a). This comparative analysis demonstrated the reliability and precision of the developed system, confirming its suitability for measuring ultra-pure water conductivity. The results validated the potential of the system to serve as an effective and economical alternative to commercially available solutions in similar applications.

### 3.2. Design and Manufacturing of the Embedded System Circuit Board

The results of the tests conducted on the prototype board confirmed that the system operates accurately. Subsequently, a professional PCB was designed and prepared for manufacturing using the EASYEDA platform (Figure 15a). After manufacturing, the PCB underwent comprehensive testing to validate its performance and reliability. The assembled system was evaluated for power delivery, signal integrity, and communication pathways, ensuring it met design specifications and performed consistently under demanding conditions.

The finalized PCB design is presented in Figure 15b. This professionally designed PCB enhances the system’s durability and reliability, making it well suited for real-world applications. The fully cased and ready-to-use version of the board is shown in Figure 15c.

### 3.3. System Operation and Analysis

The embedded system was designed to process sensor data accurately, apply calibration parameters, and present the results to the user while ensuring seamless internet connectivity for remote monitoring. Calibration values are used to adjust the raw data obtained from the sensors, such as conductivity and temperature, to ensure the accuracy of the system’s output.

The system sotware incorporates a user interface with a menu structure, allowing users to configure operational parameters. These menus provide functionalities for setting calibration constants, threshold values, and network credentials, enabling users to tailor the system to their specific application requirements. Once configured, the system continuously retrieves data from the sensors, applies the calibration adjustments, and displays the corrected values on the LCD screen. Simultaneously, the data are transmitted to a web server for storage and remote access.

Figure 16 shows the system’s LCD outputs, highlighting its key functions. In Figure 16a, water conductivity (0.05 μS) and temperature (23.90 °C) are displayed. Figure 16b shows water resistance (18.2 MΩ) and membrane pressure (0.00 bar). Cumulative water consumption in liters is shown in Figure 16c, while Figure 16d shows the device’s local IP address, confirming successful network connectivity. Finally, Figure 16e shows the system’s unique identifier number for traceability.

The calibration of the device directly depends on the ratio of the distance between the sensor electrodes and their surface area, as described in Equation (Equation 3). Even slight variations in this ratio may change for each sensor, particularly at lower values, affecting measurement accuracy. To address this, a potentiometer has been integrated into the circuit board, allowing for precise adjustment of the calibration value.

This hardware-based calibration mechanism ensures that the system can compensate for tiny variations in the sensor’s geometry, enabling consistent and accurate readings across different sensor probes. Fine-tuning the calibration using the potentiometer ensures consistent and reliable performance of the embedded system, even with sensors that have minor manufacturing variations. This method improves the system’s precision and adaptability, making it suitable for a variety of applications.

As shown in Figure 17, the $R_{REF}$ potentiometer is utilized to adjust the tolerance value of the reference resistance, ensuring accurate signal processing. Additionally, the CLEARTY potentiometer is specifically designed for calibrating the conductivity sensor.

The calibration process involves immersing the sensor in a reference liquid with a known conductivity value. The CLEARTY potentiometer is then adjusted until the measured values align with the expected reference range. Once the appropriate settings are achieved, the calibration of the conductivity sensor is complete.

### 3.4. Evaluation of the IoT-Based Water Quality Monitoring System

The data transmitted from the IoT-based monitoring system to the web server provide real-time insights into critical parameters, including membrane pressure, water resistance, water conductivity, water temperature, and water consumption. As shown in Figure 18a, the membrane pressure graph displays fluctuations in pressure over 200 samples. These variations indicate periods of stability, interspersed with spikes and drops that may reflect changes in operational conditions such as flow regulation or the performance of filtration membranes.

The water resistance graph, shown in Figure 18b, demonstrates an inverse relationship with conductivity. Fluctuations in the water resistance graph are due to changes in ionic concentration or temperature variations. In Figure 18c, water conductivity data show corresponding changes at the micro-Siemens (µS) level. The fluctuations in conductivity values are likely due to the same reasons as the water resistance values.

The water temperature graph, shown in Figure 18d, indicates a generally stable temperature range between 24 °C and 25.5 °C, with minor variations. The initial dip in temperature may represent the system’s initialization phase or transient thermal changes during operation. Lastly, cumulative water consumption, shown in Figure 18e, displays a steady ultra-pure water trend, indicating continuous flow through the system.Periods of plateaus in the graph correspond to system inactivity or reduced water usage.

To evaluate the accuracy of the developed system, a comparison was conducted against the Orentro CCT-3300 [38] conductivity sensor, which is currently utilized for ultra-pure water conductivity measurements at Sivas Numune Hospital. The measurement data obtained from both systems are presented in Table 2, highlighting the performance and reliability of the developed system.

Figure 19 presents a comparative performance between the Orentro CCT-3300 conductivity sensor and our system. Each bar corresponds to the conductivity measurements for a specific sample, with the Orentro CCT-3300 and our system displayed side by side for a direct and intuitive comparison. The chart shows a high level of concurrence between the two systems as evidenced by the nearly identical values across most samples. These results highlight the accuracy and sensitivity of our system. Minor derivations observed in a few samples fall within acceptable measurement tolerances, as detailed in Table 2.

These graphs collectively validate the system’s ability to monitor key ultra-pure water quality and operational parameters effectively. The transmitted data highlight the accuracy and responsiveness of the sensors in detecting rapid changes and maintaining reliable performance, making the system highly suitable for remote monitoring in critical applications such as healthcare and industrial water purification.

## 4. Results and Discussions

This study covers the development of a conductivity sensor probe and an embedded system circuit board capable of accurately measuring conductivity within the range of 0–100 µS/cm, a critical parameter for monitoring ultra-pure water quality. The system was designed to operate seamlessly within an IoT framework, with real-time data automatically recorded on a web server. These records provided a reliable dataset for monitoring and analysis, receiving positive feedback from both the ultra-pure water provider companies and the management of the test site, Sivas Numune Hospital. Stakeholders emphasized the system’s practical benefits in maintaining water quality standards for sensitive applications, such as dialysis centers and medical laboratories.

A number of challenges arose during the development and testing phases. It was observed that the electromagnetic noise generated by the AC excitation signal introduced significant difficulties in analog measurements. This noise not only reduced measurement resolution but also increased the complexity of data acquisition. These findings underscore the importance of advanced circuit design techniques to enhance signal integrity.

The research suggests that using a professionally fabricated PCB can greatly improve the accuracy and reliability of the system by minimizing electromagnetic interference. Moreover, replacing the current switching-mode power supply with a linear regulator for symmetric power delivery would likely reduce electromagnetic noise even further, creating a cleaner signal path and improving the precision of conductivity measurements. These adjustments would address current limitations and enhance the system’s overall performance.

The main reason for the fluctuations of the conductivity levels in ultra-pure water samples, as shown in Figure 18c, is the tendency of the substance to dissolve in water. Since the volume/surface area ratio is low in large containers, the rate of dissolution is relatively slow. However, in small containers such as pipes through which water passes, this ratio is higher and dissolution occurs faster. This is due to the ability of water to dissolve many substances, including metals, in the material of the container. When water is in a continuous flow, the rate of dissolution decreases because even if the amount of solution carried by the passing water increases, the flow of water compensates for this and reduces the intensity of the dissolution. Therefore, the continuous movement of water can be considered a limiting factor for the dissolution of the substance.

The comparative analysis between the Orentro CCT-3300 and our system demonstrates the high accuracy and reliability of the ultra-pure water conductivity measurement, with results showing strong consistency across samples. The minor deviations observed are acceptable as variance due to environmental factors within the tolerance levels. These results validate our system as a robust and reliable conductivity measurement setup.

This research provides a practical and innovative solution for real-time ultra-pure water quality monitoring while offering valuable insights for future improvements in sensor and embedded system design. By bridging the gap between laboratory precision and industrial scalability, this work paves the way for the advancement of water quality monitoring technologies in healthcare, industrial, and environmental applications.

## 5. Conclusions

In this study, a reliable and robust IoT-based ultra-pure water quality monitoring system was developed and successfully tested in a real-world setting. The system integrates a series of sensors, including a conductivity sensor, a temperature sensor, a flow meter, and a pressure sensor. These sensors are capable of monitoring and analyzing in real time a range of important water quality parameters, including conductivity, temperature, water resistance, flow rate, and water consumption. The wireless communication capabilities of the system ensure remote data access, storage, and visualization, providing significant advantages for critical applications such as dialysis centers, medical laboratories, and semiconductor manufacturers.

The accuracy and precision of the system were validated through testing at Sivas Numune Hospital, demonstrating that the system’s performance is comparable to that of conventional conductivity meters. The integration of calibration mechanisms and cloud-based data management further enhances its adaptability, ensuring reliable performance under varying conditions. By continuously monitoring fluctuations in membrane pressure, water flow, and temperature, the system provides actionable insights that ensure water quality remains within the strict standards required for sensitive medical and laboratory applications.

Future work will focus on enhancing system scalability and performance through the integration of machine learning for missing data imputation and optimizing energy efficiency for long-term deployments, and expanding the monitored parameters to include pH for a more comprehensive assessment of water quality.

## Figures and Tables

**Figure 1 sensors-25-01186-f001:**
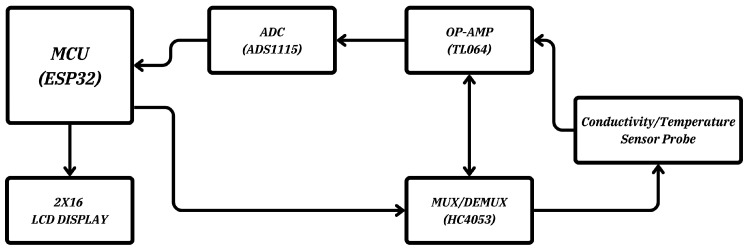
Embedded system block diagram (MCU: Microcontroller Unit (ESP32), MUX/DEMUX: Multiplexer/Demultiplexer (HC4053), OP-AMP: Operational Amplifier (TL064), ADC: analog-to-digital converter (ADS1115).

**Figure 2 sensors-25-01186-f002:**
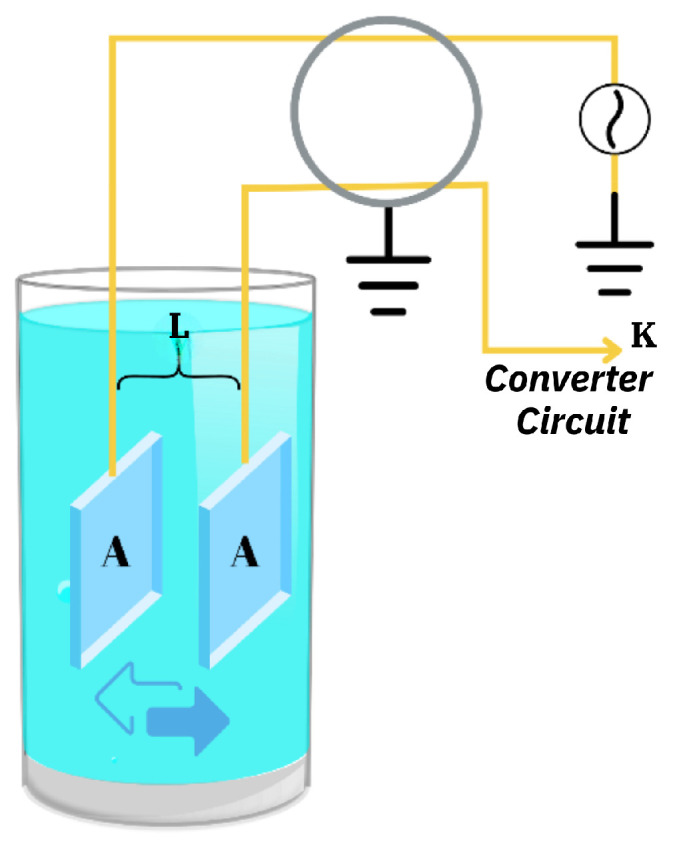
Ultra-pure water conductivity cell diagram [30].

**Figure 3 sensors-25-01186-f003:**
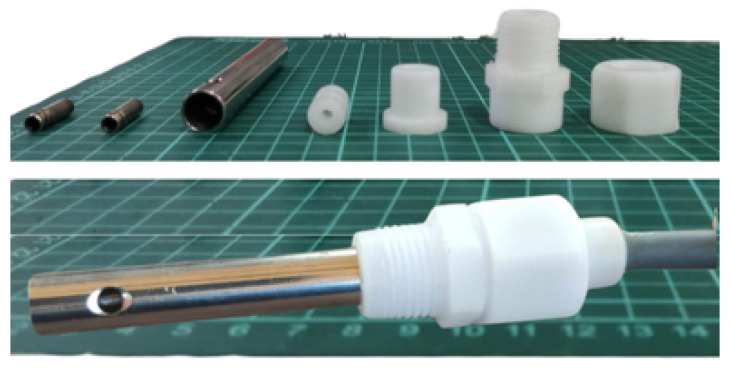
(**Top**) conductivity sensor probe parts and design. (**Bottom**) completed sensor probe.

**Figure 4 sensors-25-01186-f004:**
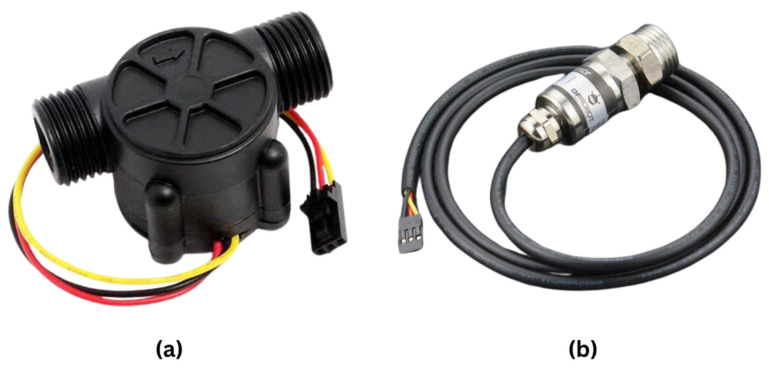
(**a**) Water flow sensor (**b**). Water pressure sensor.

**Figure 5 sensors-25-01186-f005:**
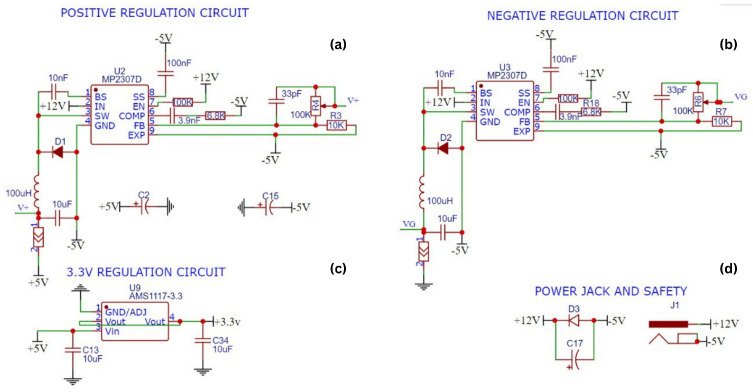
The symmetric power regulation and distribution circuits design. (**a**) Positive regulation circuit. (**b**) Negative regulation circuit. (**c**) 3.3 V regulation circuit. (**d**) Power jack and safety circuit.

**Figure 6 sensors-25-01186-f006:**
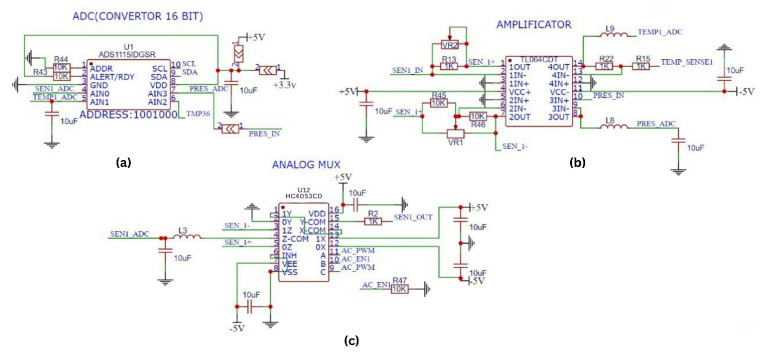
Signal processing circuit design. (**a**) ADC module for analog-to-digital conversion, (**b**) amplifier module for signal amplification, and (**c**) analog multiplexer for signal selection and routing.

**Figure 7 sensors-25-01186-f007:**
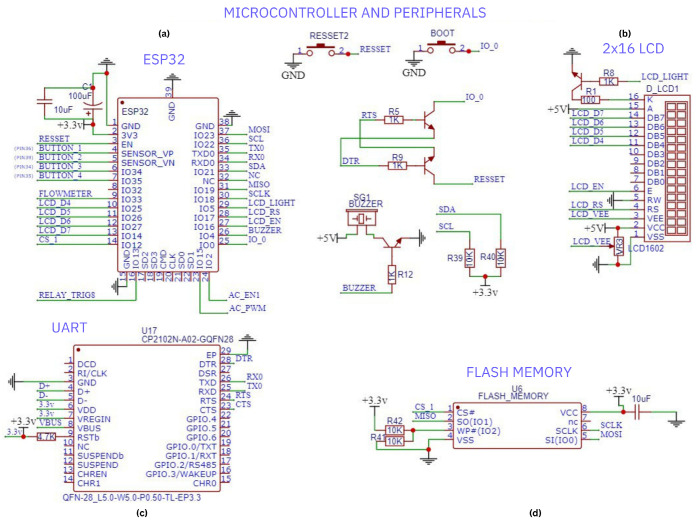
Microcontroller and peripheral connection: (**a**) ESP32 microcontroller module, (**b**) 2 × 16 LCD interface, (**c**) UART communication module, (**d**) flash memory module.

**Figure 8 sensors-25-01186-f008:**
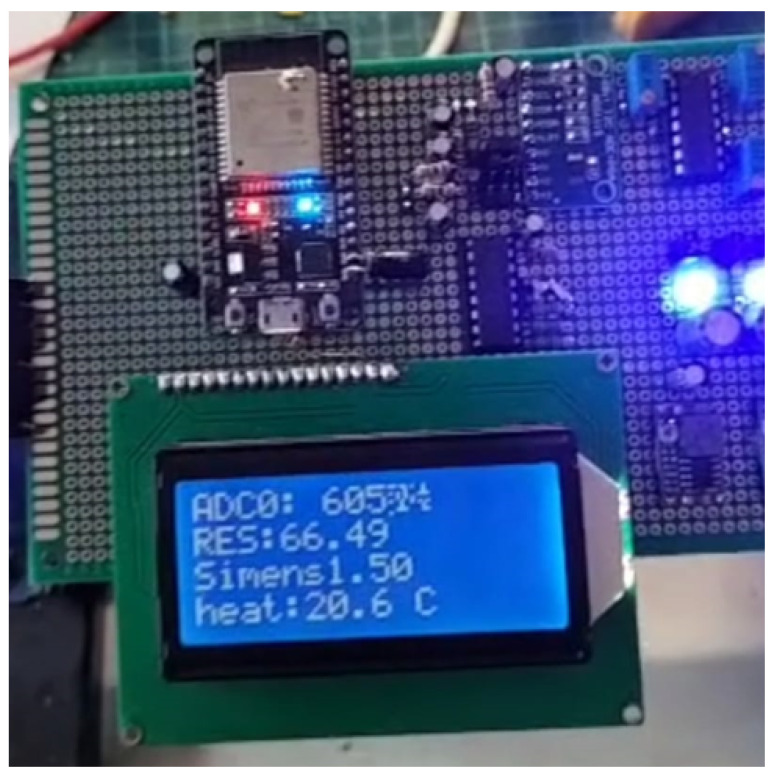
The circuit components were assembled on a perforated prototyping board.

**Figure 9 sensors-25-01186-f009:**
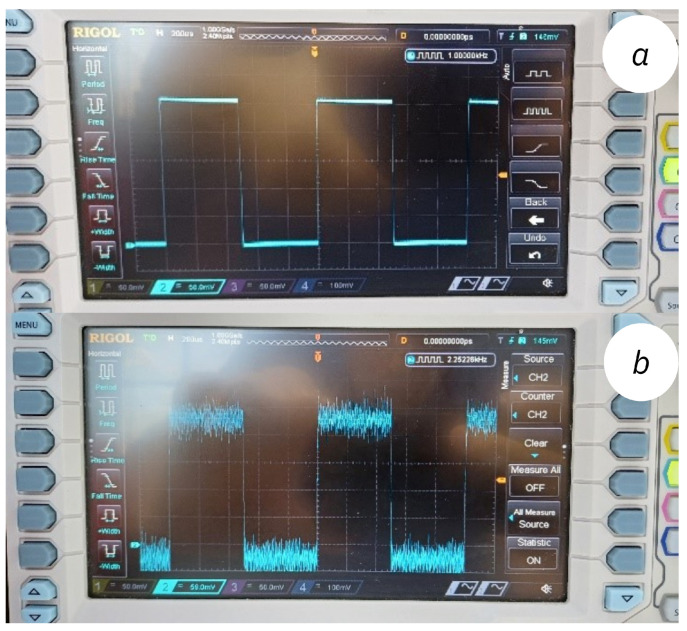
(**a**) The reference AC_PWM signal used for measurement. (**b**) The raw output signal observed during the process.

**Figure 10 sensors-25-01186-f010:**
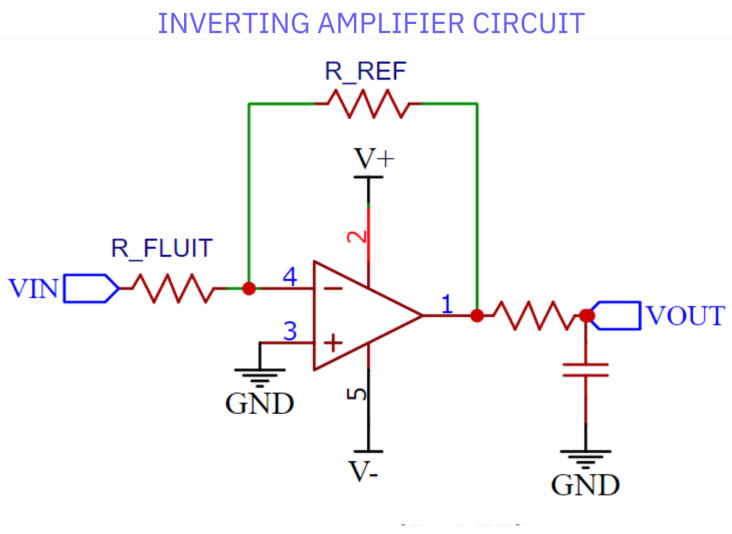
The inverting amplifier circuit.

**Figure 11 sensors-25-01186-f011:**
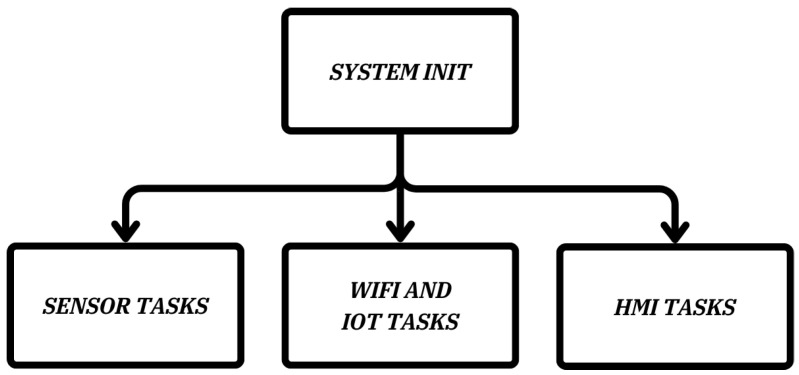
Embedded system software block diagram.

**Figure 12 sensors-25-01186-f012:**
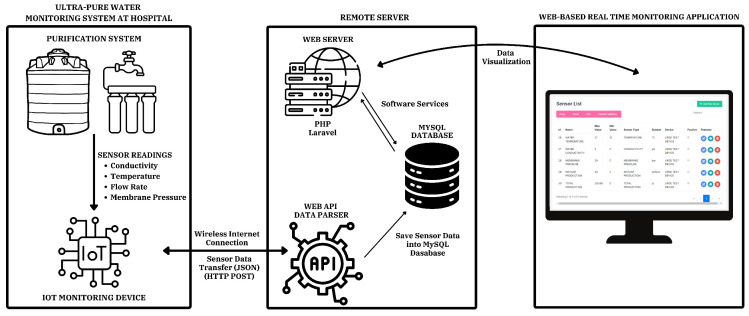
Ultra-pure water monitoring system software block diagram.

**Figure 13 sensors-25-01186-f013:**
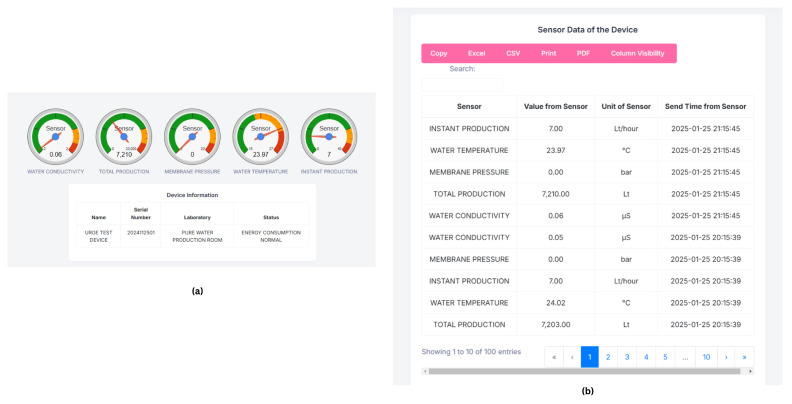
Sample of sensor monitoring web application. (**a**) Real-time sensor values. (**b**) Sensor values with 1 h intervals.

**Figure 14 sensors-25-01186-f014:**
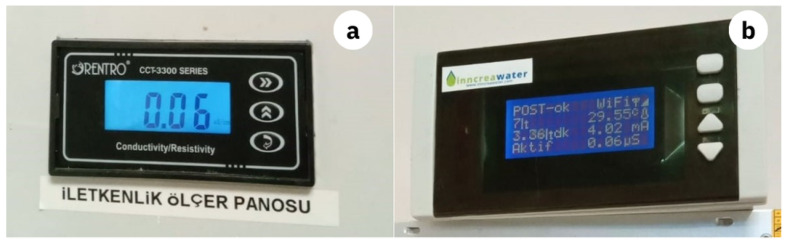
Testing the prototype. (**a**) Local conductivity sensor. (**b**) Prototype board.

**Figure 15 sensors-25-01186-f015:**
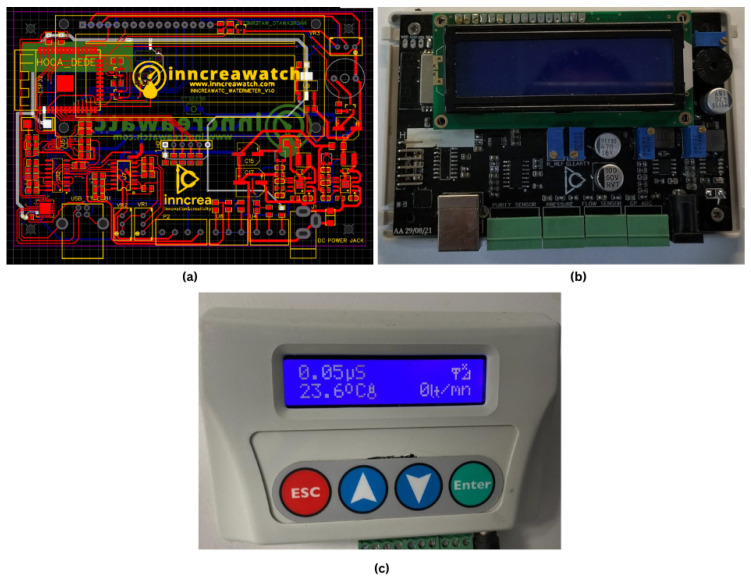
Hardware design and user interface. (**a**) PCB design layout. (**b**) Assembled PCB board. (**c**) Ready-to-use board.

**Figure 16 sensors-25-01186-f016:**
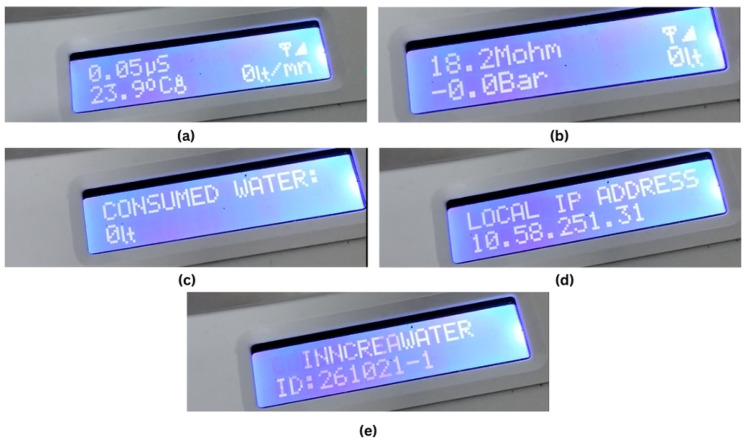
LCD screen outputs of the system. (**a**) Conductivity and temperature. (**b**) Water resistivity and pressure. (**c**) Consumed water. (**d**) Network information. (**e**) System identification.

**Figure 17 sensors-25-01186-f017:**
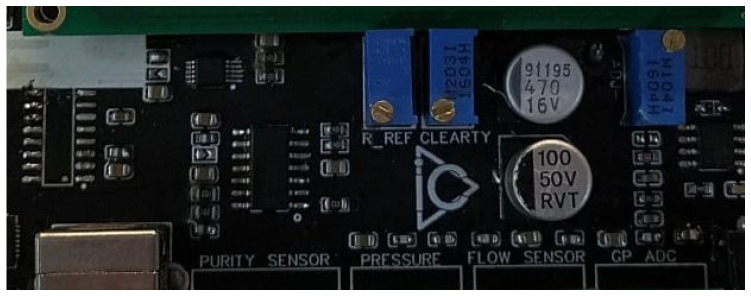
The R_REF and CLEARTY potentiometers used in the calibration process of the system.

**Figure 18 sensors-25-01186-f018:**
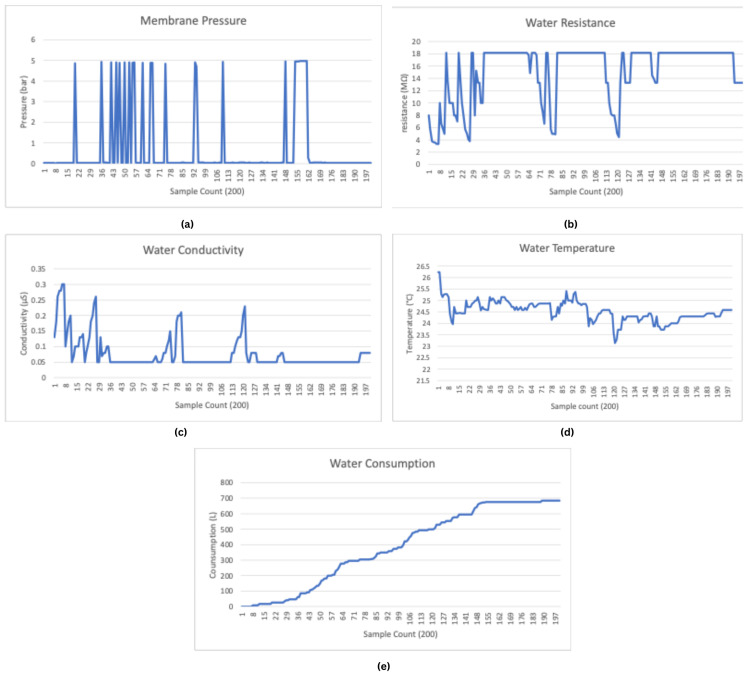
System performance data. (**a**) Membrane pressure (bar)—measured to assess system pressure fluctuations. (**b**) Water resistance (MΩ). (**c**) Water conductivity (μS). (**d**) Water temperature (°C). (**e**) Water Consumption (L).

**Figure 19 sensors-25-01186-f019:**
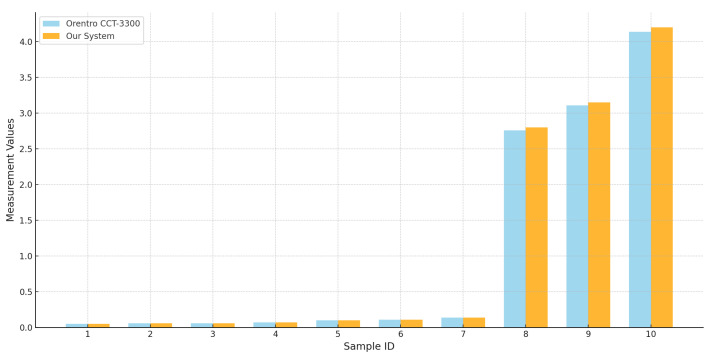
Data comparison between CCT-3300 and our system.

**Table 1 sensors-25-01186-t001:** ESP32-WROOM-32 microcontroller specifications [31].

Item	Specification
Processor	Dual-Core 32-bit LX6 microprocessor, 160 or 240 MHz
Rom	448 KB
SRam	520 KB
Rtc SRam	8 KB
Flash Memory	4 MB/16 MB
Peripherals	Wi-Fi, BLE, up to 32 GPIO, USB Bridge, ADC/DAC support
Power	3.3 V, 80 mA Active 4 μA Sleep current

**Table 2 sensors-25-01186-t002:** Comparison of conductivity values between the CCT-3300 [38] and our system.

Sample ID	CCT-3300	Our System	Margin (%)
1	0.05	0.05	0.00
2	0.06	0.06	0.00
3	0.06	0.06	0.00
4	0.07	0.07	0.00
5	0.1	0.1	0.00
6	0.11	0.11	0.00
7	0.14	0.14	0.00
8	2.76	2.8	1.45
9	3.11	3.15	1.29
10	4.14	4.2	1.45

## Data Availability

Data is contained within the article.

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
