# Peer review of "Development of an Internet of Things-Based Ultra-Pure Water Quality Monitoring System"

_sensors, 2025, doi:10.3390/s25041186_

Round 1
Reviewer 1 Report
Comments and Suggestions for Authors
The topic of the paper is of interest. It presents an IoT-based ultra-pure water quality monitoring system.
Below are my comments:
- Only 3 related works are summary presented. In my opinion several more papers should be presented and their obtained quantitative results/disadvantages should be discussed.
- Regarding Figure 1, it would be good to contain a legend explaining all the acronyms, considering that they were not presented in the text.
- Please provide a reference for the technical specification presented in Table 1.
- Please provide references for all hardware components (ADS115, HC4053, MP2307, etc.).
- Please define the acronyms at their first use, even though there is a list of abbreviations.
- In my opinion more details about the developed web application should be provided in section 2.7. I suggest to replace/modify Figure 12 in order to present the particularities you have implemented (i.e., a sample of the JSON file, interfaces of the web application).
- There are some paragraphs where there is no space between the value and the unit of measurement (i.e., section 2.3). In line 350 there is no whitespace between meter and Figure 13b. Also, in line 358 there is a dot before Figure 14a. Please check all the sections and correct all the typos mistakes.
Author Response
For Development of an Internet of Things-Based Ultra-pure Water Quality Monitoring System
Dear Reviewer,
We agree with your valuable comments and have reorganized the article in line with your feedback. All revisions made on the article were highlighted on the pdf file (revisions_highlighted.pdf) and uploaded to the system in the supplementary files section. Color codes are created for the labeling of the revisions made. Cyan was used for Reviewer 1, green for Reviewer 2 and yellow for common revisions. The responses to your comments are given below respectively.
We hope our response to your comments address your concerns about the study. Thank you for your time and detailed review.
Comment 1: Only 3 related works are summary presented. In my opinion several more papers should be presented, and their obtained quantitative results/disadvantages should be discussed.
Response 1: We have expanded the Introduction section by incorporating three additional research papers that are related to semiconductor industry. (Between line numbers 22 and 27, reference numbers 13, 14, 15 and 16). Also, another related works is added with reference numbers 27, 28, 29. From line 51 to 75 reorganized and detailed.
Comment 2: Regarding Figure 1, it would be good to contain a legend explaining all the acronyms, considering that they were not presented in the text.
Response 2: We have updated the label of Figure 1 to include a comprehensive legend that explains all the acronyms used in the figure. Also, some misspellings are corrected.
Comment 3: Please provide a reference for the technical specification presented in Table 1.
Response 3: The technical specifications presented in Table 1 are referenced (reference number 31).
Comment 4: Please provide references for all hardware components (ADS115, HC4053, MP2307, etc.).
Response 4: References for all hardware components mentioned, including ADS115, TL064, HC4053, MP2307, YF-S201, Analog water pressure sensor and Orentro-3300 are added to the manuscript as reference number 32, 33, 34, 35, 36, 37 and 38, respectively.
Comment 5: Please define the acronyms at their first use, even though there is a list of abbreviations. 
Response 5: All acronyms have been defined upon their first use in the text, even though a comprehensive list of abbreviations is provided. Ultra-pure water no longer abbreviated.
Comment 6: In my opinion more details about the developed web application should be provided in section 2.7. I suggest to replace/modify Figure 12 in order to present the particularities you have implemented (i.e., a sample of the JSON file, interfaces of the web application).
Response 6: Figure 12 has been updated to provide more detailed information about the developed web application. The updated figure now includes the interface of the web application, showcasing its functionality and design, as well as a diagram of the ultra-pure water monitoring system. Also, in order to support Figure 12, Figure 13 which provides information about sample of sensor monitoring web application is added. From line number 379 to 383 an explanation is added for Figure 13.
Comment 7: There are some paragraphs where there is no space between the value and the unit of measurement (i.e., section 2.3).  In line 350 there is no whitespace between meter and Figure 13b. Also, in line 358 there is a dot before Figure 14a. Please check all the sections and correct all the typos mistakes.
Response 7: A review of the manuscript is conducted to identify and correct all typographical errors, including those noted in your comment. Specifically, spaces have been added between values and their units of measurement throughout the text, ensuring consistency and adherence to standard formatting guidelines.

Reviewer 2 Report
Comments and Suggestions for Authors
Detailed Comments to Authors
1. The authors clearly state the problem they are trying to solve; however, their review of existing works does not identify the measurement of ultra-pure water or IoT framework as a research gap.
2. The authors should review their mathematical formulations. For example, they refer to (2) (4) as conductivity instead of conductance and they have used the conductivity term throughout the paper. It makes the reader wonder what quantity they are really referring to.
3. The authors would have to justify why (4), determined by measuring current and voltage through the liquid, differs from (5).
4. Figures 5 and 6 caption refers to subfigures (a),(b), …, but are missing in the figures. The figures should be labeled appropriately.
5. How the sensor probe is interfaced with the rest of the system is not clearly shown. SEN1_IN, SEN 1+ and SEN1- should be defined.
6. Considering that the probes deal with very low currents, the authors should show how they dealt with the probe noise so it does not affect the system’s accuracy.
7. The authors show various results captured in Figure 17. These results are not benchmarked against any measurement or lab results and are therefore not sufficient to conclude that the system is reading accurate results.
Author Response
For Development of an Internet of Things-Based Ultra-pure Water Quality Monitoring System
Dear Reviewer,
We agree with your valuable comments and have reorganized the article in line with your feedback. All revisions made on the article were highlighted on the pdf file (revisions_highlighted.pdf) and uploaded to the system in the supplementary files section. Color codes are created for the labeling of the revisions made. Cyan was used for Reviewer 1, green for Reviewer 2 and yellow for common revisions. The responses to your comments are given below respectively.
We hope our response to your comments address your concerns about the study. Thank you for your time and detailed review.
Comment 1: The authors clearly state the problem they are trying to solve; however, their review of existing works does not identify the measurement of ultra-pure water or IoT framework as a research gap.
Response 1: We have expanded the Introduction section by incorporating three additional research papers that are related to semiconductor industry. (Between line numbers 22 and 27, reference numbers 13, 14, 15 and 16). Also, another related works are added with reference numbers 27, 28, 29. From line 51 to 75 reorganized and detailed.
Comment 2: The authors should review their mathematical formulations. For example, they refer to (2) (4) as conductivity instead of conductance and they have used the conductivity term throughout the paper. It makes the reader wonder what quantity they are really referring to.
Response 2: The terminology used in mathematical formulations has been carefully reviewed and revised to eliminate any confusion between conductance and conductivity. Specifically, the incorrect references to "conductivity" in equations (2) and (4) have been corrected to "conductance," where applicable. Additionally, detailed explanations of these formulas have been updated in accordance with the relevant references to ensure accuracy and clarity.
Comment 3: The authors would have to justify why (4), determined by measuring current and voltage through the liquid, differs from (5).
Response 3: An explanation has been added before Equation (5) to clarify why it differs from Equation (4). This explanation highlights the distinctions between the two equations, specifically addressing the methods used to derive them and the underlying assumptions.
Comment 4: Figures 5 and 6 caption refers to subfigures (a), (b), …, but are missing in the figures. The figures should be labeled appropriately.
Response 4: The captions of Figures 5 and 6 have been updated to include labels for the subfigures as referenced in the text. The figures themselves have also been appropriately labeled to correspond with the captions, ensuring consistency and clarity.
Comment 5: How the sensor probe is interfaced with the rest of the system is not clearly shown. SEN1_IN, SEN 1+ and SEN1- should be defined.
Response 5: To address this, the interfacing of the sensor probe with the rest of the system has been clarified in the text. Specifically, from lines 252 to 261, detailed explanations of the SEN1_IN, SEN1+, and SEN1- signals have been added, including their roles and how they interact within the system.
Comment 6: Considering that the probes deal with very low currents, the authors should show how they dealt with the probe noise, so it does not affect the system’s accuracy.
Response 6: A detailed explanation has been added from lines 267 to 273, describing the methods employed to manage probe noise and minimize its impact on the system's accuracy. This includes the noise reduction techniques utilized, such as signal filtering, shielding, and calibration procedures, ensuring reliable measurements despite the low current levels involved.
Comment 7: The authors show various results captured in Figure 17. These results are not benchmarked against any measurement or lab results and are therefore not sufficient to conclude that the system is reading accurate results.
Response 7: The various results present in Figure 17 (after revision it is Figure 18) is supported with Table 2 and Figure 19. The measurement data obtained from both systems are presented in Table 2 highlighting the performance and reliability of the developed system. Figure 19 presents a comparative performance between the Orentro CCT-3300 conductivity sensor and our system. And also a new discussion paragraph is added into Result and Discussion section between line numbers 516 and 520.

Reviewer 3 Report
Comments and Suggestions for Authors
The abstract, introduction and overall paper is very well organized and well written.
The missing parts of literature are clearly identified in section 1.
Section 2 describes the overall design and technical details of the system presented in the study. Again, it is very well described.
There are mainly two contributions of this study.
1. The development of a conductivity sensor probe, and
2. An embedded system circuit board capable of accurately measuring conductivity
I find some lack of scientific contribution from the paper. I would also like to raise the question of novelty in this paper. The only novel contribution in this paper is the development of the conductivity sensor probe.
The scientific language used in the paper is excellent. The organization of the paper is excellent.
Author Response
For Development of an Internet of Things-Based Ultra-pure Water Quality Monitoring System
Dear Reviewer,
Thank you for your constructive feedback and positive comments on the organization and language of the manuscript. All revisions made on the article were highlighted on the pdf file (revisions_highlighted.pdf) and uploaded to the system in the supplementary files section. Color codes are created for the labeling of the revisions made. Cyan was used for Reviewer 1, green for Reviewer 2 and yellow for common revisions. We acknowledge your comments about the scientific contribution and novelty of the study and would like to address these points in detail.
Comment1: I find some lack of scientific contribution from the paper. I would also like to raise the question of novelty in this paper. The only novel contribution in this paper is the development of the conductivity sensor probe.
Response1:
- Scientific Contribution
- This paper presents not only the development of a conductivity sensor probe, but also a comprehensive embedded IoT-based system specifically designed for ultra-pure water (UPW) quality monitoring. The integration of a conductivity sensor, an embedded system built for the sensor, and an IoT-based device monitoring web application makes it possible to perform real-time conductivity monitoring for highly sensitive applications such as medical laboratories, dialysis centers, and semiconductor manufacturing. The IoT-based UPW system we present in this paper is differentiated by features that are not available in existing conventional pure water monitoring systems.
- PCB board and circuit designs specifically developed for ultrapure water conductivity measurement provide robust measurement accuracy. For example, the developed system is able to overcome challenges such as electromagnetic interference and temperature compensation, which are critical for precise conductivity measurements at micro siemens per centimeter (µS/cm) levels.
- Novelty
- While the development of the conductivity sensor probe is highlighted as a key contribution, the novelty of the study also lies in the integration of hardware and software components, ensuring real-time, scalable, and reliable performance.
- The embedded system circuit board adapted for precise calibration of the sensor probe aimed to provide adaptability to different conditions while maintaining accuracy.
- The inclusion of IoT data storage and visualization facilitates continuous monitoring, historical data analysis, and real-time alerts, which are not commonly addressed in existing studies.
- Unlike traditional monitoring systems that rely on manual or periodic measurements, the proposed system automatically records, stores, and visualizes conductivity, temperature, flow rate, and pressure data in real time, and integrate them into a user-friendly interface.
- Impact
- The system has been validated under real-world conditions at Sivas Numune Hospital, demonstrating its applicability in critical healthcare environments. This practical deployment and positive feedback from stakeholders further underscore the relevance and contribution of this work to both academia and industry.
- Its deployment in a hospital laboratory validates its practical effectiveness, ensuring compliance with stringent water quality standards for medical and diagnostic procedures.
- Unlike traditional ultra-pure water monitoring systems, this study addresses a clear gap in the literature by providing an IoT-based and real-time system design and implementation example for ultrapure water monitoring.
We hope these clarifications address your concerns regarding the scientific contribution and novelty of the study. If there are additional areas you would like us to elaborate on or revise, we would be happy to make further adjustments. The file titled “revisions_highlighted.pdf” containing the requested changes from all reviewers has been added to the “Reply to Review Report” section as a supplementary file. Thank you for your time and detailed review.

Round 2
Reviewer 1 Report
Comments and Suggestions for Authors
Thank you to the authors for taking all my suggestions into account. I believe that the current form of the article is one that can be published in the journal. Accordingly, I recommend the paper to be accepted in this form.
Author Response
Dear Reviewer,
We sincerely appreciate your thoughtful feedback and the time you dedicated to reviewing our manuscript. Your valuable suggestions have significantly contributed to improving the quality of our work. We are delighted to hear that you find the revised version suitable for publication.
The final revised and highlighted version of the manuscript is enclosed.
Thank you for your recommendation and for your constructive input throughout the review process.
Best regards,

Reviewer 2 Report
Comments and Suggestions for Authors
The Authors have amply revised the manuscript. There are some minor issues that must be addressed to improve the readability of the paper.
The minor comments are below:
Line 113 should be corrected to read as: The measured conductance, on the other hand, is the reciprocal of resistance (R) and can …
The confusion between conductance and conductivity still exists a per the statement in line 116-117 and equation 4. The recommendation is for authors to use standard symbols to avoid confusing the readers
Author Response
Dear Reviewer,
We sincerely appreciate your time and effort in reviewing our manuscript and for providing valuable feedback to further enhance its readability. Below are our responses to your comments:
Comment 1: Line 113 should be corrected to read as: The measured conductance, on the other hand, is the reciprocal of resistance (R) and can …
The confusion between conductance and conductivity still exists a per the statement in line 116-117 and equation 4. The recommendation is for authors to use standard symbols to avoid confusing the readers
Response 1: We understand the confusion and have carefully revised the text in lines 113, 115-117 and equation 4 to ensure a clear distinction between conductance and conductivity. In addition, we have highlighted all revisions in the pdf file titled manuscript_rev2_highlighted. Moreover, labels of Figure 12 and Figure 19 are updated for better understanding.
The final revised and highlighted version of the manuscript is enclosed.
Thank you for your recommendation and for your constructive input throughout the review process.
Best regards,
